# Embedding a random graph via GNN: mean-field inference theory and RL applications to NP-Hard multi-robot/machine scheduling

## Abstract

We develop a theory for embedding a random graph using graph neural networks (GNN) and illustrate its capability to solve NP-hard scheduling problems. We apply the theory to address the challenge of developing a near-optimal learning algorithm to solve the NP-hard problem of scheduling multiple robots/machines with time-varying rewards. In particular, we consider a class of reward collection problems called Multi-Robot Reward Collection (MRRC). Such MRRC problems well model ride-sharing, pickup-and-delivery, and a variety of related problems. We consider the classic identical parallel machine scheduling problem (IPMS) in the Appendix.

For the theory, we first observe that MRRC system state can be represented as an extension of probabilistic graphical models (PGMs), which we refer to as random PGMs. We then develop a mean-field inference method for random PGMs. We prove that a simple modification of a typical GNN embedding is sufficient to embed a random graph even when the edge presence probabilities are interdependent.

Our theory enables a two-step hierarchical inference for precise and transferable Q-function estimation for MRRC and IPMS. For scalable computation, we show that transferability of Q-function estimation enables us to design a polynomial time algorithm with $1 - 1/e$ optimality bound. Experimental results on solving NP-hard MRRC problems (and IMPS in the Appendix) highlight the near-optimality and transferability of the proposed methods.

## 1 Introduction

Consider a set of identical robots seeking to serve a set of spatially distributed tasks. Each task is given an initial age (which then increases linearly in time). Greater rewards are given to younger tasks when service is complete according to a predetermined reward rule. We focus on NP-hard *scheduling* problems possessing constraints such as 'no possibility of two robots assigned to a task at once'. Such problems prevail in operations research, e.g., dispatching vehicles to deliver customers in a city or scheduling machines in a factory. Impossibility results in asynchronous communication[1] [Fischer et al. (1985)] make these problems inherently centralized.

**Learning-based scheduling methods for single-robot NP-hard problems.** *structure2vec* (Dai et al. (2016)) is a popular Graphical Neural Network (GNN) derived from the fixed point iteration of PGM based mean-field inference. Recently, Dai et al. (2017) showed that structure2vec can construct a solution for Traveling Salesman Problem (TSP). A partial solution to TSP was considered as an intermediate state, and the state was represented using a heuristically constructed probabilistic graphical model (PGM). This GNN was used to infer the $Q$-function, which they exploit to select the next assignment. While their choice of PGM was entirely heuristic, their approach achieved near-optimality and transferability of their trained single-robot scheduling algorithm to new single-robot scheduling problems with an unseen number of tasks. Those successes were restricted to single-robot problems except for special cases when the problem can be modeled as a variant of single-robot TSP via multiple successive journeys of a single robot, c.f., (Nazari et al. (2018); Kool et al. (2018)).

---

[1]Due to this limitation, multi-agent (decentralized) methods are rarely used in industries (e.g., factories).

**Proposed methods and contributions.** The present paper explores the possibility of near-optimally solving multi-robot, multi-task NP-hard scheduling problems with time-dependent rewards using a learning-based algorithm. This is achieved by first extending the probabilistic graphical model (PGM)-based mean-field inference theory in Dai et al. (2016) for *random* PGM. We next consider a seemingly naive two-step heuristic: (i) approximate each edge's presence probability and (ii) apply a typical GNN encoder with probabilistic adjustments. We subsequently provide theoretical results that justify this approach. We call structure2vec (Dai et al. (2016)) combined with this heuristic as *random structure2vec*. After observing that each state of a robot scheduling problem can be represented as a *random* PGM, we use random structure2vec to design a transferable (to different size problems) reinforcement learning method that is, to the best of our knowledge, the first to learn near-optimal NP-hard multi-robot/machine scheduling with time-dependent rewards. Experiments yield $97\%$ optimality for MRRC problems in a deterministic environment with linearly-varying rewards. This performance is well extended to experiments with stochastic traveling time.

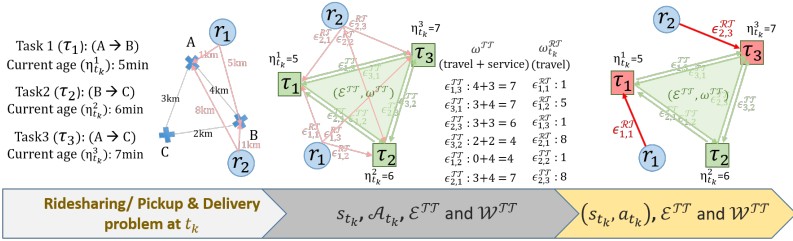

Figure 1: Representing a ridesharing/pickup and delivery problem as an MRRC problem

## 2 MULTI-ROBOT SCHEDULING PROBLEM FORMULATION

We formulate a multi-robot reward collection problem (MRRC) as a discrete-time, discrete-state (DTDS) sequential decision-making problem. (For a closely related continuous-time and continuous-state (CTCS) formulation of IPMS problems, see Appendix A.1). For the DTDS formulation, time advances in fixed increments $\Delta$ and each such time-step is considered a decision epoch. (For the CTCS formulation, the times when a robot arrives at a task or completes a task are considered as decision epochs.) At such decision epochs (which occur at every time-step in our discrete-time model), we reassign all available robots to remaining tasks. We use $k$ to index the decision epochs and let $t_k$ denote the time that epoch $k$ occurs (in discrete-time $t_k = k \cdot \Delta$). We assume that at each decision epoch, we are newly given the duration of time required for a robot to complete each task, which we call *task completion time*. Such task completion times may be constants or random variables, and in either case, they are determined by current state (e.g., locations of the robot and the task) at each epoch.[2] We consider initial tasks as nodes in a fully connected graph. For the edge from task $p$ to task $q$, we denote as $\epsilon_{p,q}^{TT}$. The edge weight assigned is the task completion time for a robot that has just completed task $p$ to subsequently complete task $q$. Let $\mathcal{E}^{TT}, \mathcal{W}^{TT}$ be the set of all $\epsilon_{p,q}^{TT}$ and the set of corresponding weights. All elements of $\mathcal{W}^{TT}$ are multiples of $\Delta$).

**State.** The state $s_{t_k}$ at time $t_k$ is represented as $\left(\mathcal{G}_{t_k}, \mathcal{W}_{t_k}^{RT}, \alpha_{t_k}\right)$. $\mathcal{G}_t$ is a directed bipartite graph $(\mathcal{R} \cup \mathcal{T}_{t_k}, \mathcal{E}_{t_k}^{RT})$ where $\mathcal{R}$ is the set of all robots, $\mathcal{T}_{t_k}$ is the set of all remaining unserved tasks at time step $t_k$. The set $\mathcal{E}_{t_k}^{RT}$ consists of all directed edges from robots to unserved tasks at time $t_k$. To each edge is associated a weight equal to the task completion time. Let $\mathcal{W}_{t_k}^{RT}$ denote the set of all such weights for all edges at $t_k$ (either constants or random variables, of which values are restricted to multiples of $\Delta$ in the the DTDS system). For example, $\epsilon_{i,p}^{RT} \in \mathcal{E}_{t_k}^{RT}$ is an edge indicating robot $i$ is assigned to serve task $p$. To this edge a task completion time is assigned according to current locations of the robot $i$ and task $p$. Each task is given an initial age which increases linearly with time (a multiple of $\Delta$ for DTDS). Let $\alpha_{t_k} = \{\eta_{t_k}^p \in \mathbb{R} | p \in \mathcal{T}_{t_k}\}$ denote the set of ages where $\eta_{t_k}^p$ indicates the age of task $p$ at time-step $t_k$. We denote the set of possible states as $\mathcal{S}$.

It is intuitively clear how MRRC can directly model problems with stationary tasks. MRRC can also model problems such as ride-sharing or package delivery problems in which the robot location at

---

[2]In the later case, our method only requires samples of the random variables; distributions are not required.

the start of the task is different than at the end. Consider pickup and delivery tasks as illustrated in Figure 1. Task 1, denoted as $\tau_1$, is to pickup from $A$ and deliver to location $B$. The weight assigned to the edge $\epsilon_{2,1}^{TT}$ is the task completion time for a robot who has just completed task 2, and is thus located at C, who subsequently completes task 1. The traveling distance to task 1 ($C \to A$) is 4 and the delivery distance ($A \to B$) is 3, so the task completion time is $\epsilon_{2,1}^{TT} = 3 + 4 = 7$. In the middle image in Figure 1, state $s_{t_k}$ (robots nodes, task nodes, arcs from robots to tasks and their weights, and ages), the system $\mathcal{E}^{TT}$ (arcs between task nodes) and their weights $\mathcal{W}^{TT}$ are depicted.

**Joint assignment.** Once a robot has reached a task, it will conduct it until completion. Otherwise, we allow reassignment prior to arrival. Thus, available robots can change their assignments whenever a decision epoch occurs. A joint assignment of robots to tasks at current state $s_{t_k} = \left(\mathcal{G}_{t_k}, \mathcal{W}_{t_k}^{RT}, \alpha_{t_k}\right)$, denoted as $a_{t_k}$, should satisfy: (i) no two robots can be assigned to the same task, and (ii) a robot may only remain without assignment when the number of robots exceeds the number of remaining tasks. Thus, a joint assignment $a_{t_k}$ is the set of edges in a maximal bipartite matching of the bipartite graph $\mathcal{G}_{t_k}$. The action space $\mathcal{A}_{t_k}$ is depends upon $s_{t_k}$, as it is defined as the set of all maximal bipartite matchings in $\mathcal{G}_{t_k}$. A policy $\pi$ is defined as $\pi(s_{t_k}) = a_{t_k}$, where $s_{t_k} \in \mathcal{S}$ and $a_{t_k} \in \mathcal{A}_{t_k}$.

**Transition function and reward.** In the hierarchical control literature, our *assignment* is termed a *macro-action*. In pursuit of the macro-action, robots may make multiple sequential *micro-actions* to serve the task. The transition probability associated with a macro-action is derived from the transition probabilities associated with micro-actions [Omidshafiei et al. (2017)]. For a joint macro-action, assume there is an induced joint micro-action denoted as $u_t \in \mathcal{U}$ with associated transition probabilities $P\left(s_{t+1}|s_t, u_t\right) : S_t \times \mathcal{U}_t \times S_t \to [0,1]$. Omidshafiei et al. (2017) proves we can calculate the corresponding 'extended transition function' $P'\left(s_{t_{k+1}}|s_{t_k}, a_{t_k}\right) : \mathcal{S}_{t_k} \times \mathcal{A}_{t_k} \times \mathcal{S}_{t_k} \to [0,1]$.

When a task is served, a reward is given according to a predetermined reward function that computes rewards according to the task's age at the time of service. Note that the state and assignment information $s_{t_k}$, $a_{t_k}$ and $s_{t_{k+1}}$ are thus sufficient to determine the reward at decision epoch $t_{k+1}$. As such we denote the reward function as $R(s_{t_k}, a_{t_k}, s_{t_k+1}) : \mathcal{S}_{t_k} \times \mathcal{A}_{t_k} \times \mathcal{S}_{t_k} \mapsto \mathbb{R}$.

**Objective.** Given an initial state $s_{t_0} \in \mathcal{S}$, the MRRC seeks to maximize the sum of expected rewards through time by optimizing an assignment policy $\pi^*$ as $\pi^* = \underset{\pi}{\operatorname{argmax}} \, \mathbb{E}_{\pi, P'}\left[\sum_{k=0}^{\infty} R\left(s_{t_k}, \pi(s_{t_k}), s_{t_{k+1}}\right)|s_{t_0}\right]$.

## 3 Scheduling by Inferencing with a Random PGM

### 3.1 Background on probabilistic graphical model (PGM) and structure2vec

**PGM.** Given random variables $\mathcal{X} = \{X_k\}$, suppose that we can factor the joint distribution $p(\mathcal{X})$ as $p(\mathcal{X}) = \frac{1}{Z} \prod_i \phi_i(\mathcal{D}_i)$ where $\phi_i(\mathcal{D}_i)$ denotes a marginal distribution or conditional distribution associated with a set of random variables $\mathcal{D}_i$; $Z$ is a normalizing constant. Then $\{X_k\}$ is called a probabilistic graphical model (PGM). In a PGM, $\mathcal{D}_i$ is called a clique and $\phi_i(\mathcal{D}_i)$ is called a clique potential for $\mathcal{D}_i$. When we write simply $\phi_i$, suppressing $\mathcal{D}_i$, $\mathcal{D}_i$ is called the scope of $\phi_i$.

**Mean-field inference with PGM.** A popular use of PGM is PGM-based mean-field inference. Suppose that $\mathcal{X} = \{\{Y_k\}, \{H_j\}\}$, where we are interested in the inference of $\{H_j\}$ given $\{Y_k\}$. For the inference problem, our interest will be calculating $p(\{H_j = h_j\} \mid \{Y_k = y_k\})$ but the calculation might not be tractable. In mean-field inference, we instead find a surrogate distribution $q(\{H_j = h_j\}) = \prod_j q_j(h_j)$ with smallest Kullback-Leibler distance to $p(\{H_j = h_j\} \mid \{Y_k = y_k\})$. This surrogate distribution is then used to conduct the inference. Hereafter, for convenience, we suppress explicit mention of the random variable, for example, we write $p(h_j)$ for $p(H_j = h_j)$. Koller & Friedman (2009) shows that when we are given a PGM, the $q(\{h_j\})$ can be obtained by a fixed point equation. Despite the usefulness of this approach, we are not often directly given the PGM.

***Structure2vec.*** In some problems such as molecule classification problems, data is given as graphs. For such special cases, [Dai et al. (2016)] suggests that this graph structure information may be enough to conduct a mean-field inference when combined with Graph Neural Network (GNN). Let us first embed $p(h_j \mid \{y_k\})$ to a vector $\tilde{\mu}_j$ using the equation $\tilde{\mu}_j = \int_{\mathcal{H}} \phi(h_j) p(h_j \mid \{y_k\}) \, dh_j$. Suppose that our problem has a special PGM structure that joint distribution is proportional to some factorization $\prod_{k \in \mathcal{V}} \phi(h_k, y_k) \prod_{i,j \in \mathcal{V}} \phi(h_i, h_j)$, where $\mathcal{V}$ denotes the set of vertex indexes. Then

according to [Dai et al. (2016)], the embedding of the fixed point iteration operation of PGM-based mean-field inference corresponds to a neural network operation $\tilde{\mu}_i = \sigma(W_1 y_i + W_2 \sum_{j \neq i} \tilde{\mu}_j)$ ($\sigma$ denotes Relu function and $W$ denotes parameters of neural networks). We can therefore use $\{\tilde{\mu}_k\}$ to solve the original inference problem instead of $p(\{h_k\} \mid \{y_j\})$ or $q(\{h_k\})$. Note that their suggested neural network operation is similar to the network structure of Graph Convolutional Networks [Kipf & Welling (2017)], a popular GNN-based graph embedding method. This observation enables one to interpret GNN-based graph embedding methods as mean-field inference using PGM.

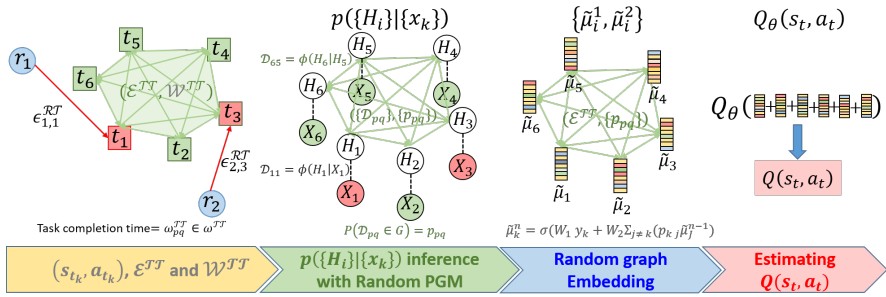

Figure 2: State representation and main inference procedure

## 3.2 INFERENCE WITH RANDOM PGM AND RANDOM GRAPH EMBEDDING WITH GNN

**Random PGM.** Denote the set of all random variables in the inference problem as $\mathcal{X} = \{X_i\}$. Suppose that the set of all possible PGMs on $\mathcal{X}$, denoted as $\mathcal{G}_{\mathcal{X}}$, is prior knowledge (e.g., for a robot scheduling problem, PGM is often a specific Bayesian Network - see Appendix A.2). A random PGM on $\mathcal{X}$ is then defined as $\{\mathcal{G}_{\mathcal{X}}, \mathcal{P}\}$ where $\mathcal{P} : \mathcal{G}_{\mathcal{X}} \mapsto [0, 1]$ is the probability measure for a realization of an element of $\mathcal{G}_{\mathcal{X}}$. Note that the inference of $\mathcal{P}$ will be difficult. To avoid this task, we start by defining *semi-cliques*. Suppose that we are given the set of all possible cliques on $\mathcal{X}$ as $\mathfrak{C}_{\mathcal{X}}$. Only a few cliques in $\mathfrak{C}_{\mathcal{X}}$ will be actually realized as an element of PGM according to $\mathcal{P}$ and become real cliques. So we call the elements $\mathcal{D}_m \in \mathfrak{C}_{\mathcal{X}}$ as *semi-cliques*. Note that if we are given $\mathcal{P}$ then we can easily calculate the presence probability $p_m$ of semi-clique $\mathcal{D}_m$ as $p_m = \sum_{G \in \mathcal{G}_{\mathcal{X}}} \mathcal{P}(G) 1_{\mathcal{D}_m \in G}$.

**Mean-field inference with random PGM.** The following theorem extends mean-field inference with PGM (Koller & Friedman (2009)) to mean-field inference with random PGM. It shows that we only need to infer the presence probability of each semi-clique in the random PGM, not $\mathcal{P}$.

*Theorem 1. Random PGM based mean field inference. Suppose we are given a random PGM on $\mathcal{X} = \{X_k\}$. Also, assume that we know presence probability $\{p_m\}$ for all semi-cliques $\mathcal{D}_m \in \mathfrak{C}_{\mathcal{X}}$. The surrogate distribution $\{q_k(x_k)\}$ in mean-field inference is locally optimal only if $q_k(x_k) = \frac{1}{Z_k} \exp\left\{\sum_{m:X_k \in \mathcal{D}_m} p_m \mathbb{E}_{(\mathcal{D}_m - \{X_k\}) \sim q}\left[\ln \phi_m(\mathcal{D}_m, x_k)\right]\right\}$ where $Z_k$ is a normalizing constant and $\phi_m$ is the clique potential for clique $m$. (For the proof see Appendix A.3.)*

**Random *structure2vec*.** From *Theorem 1*, we can develop a *random structure2vec* corresponding to a random PGM with $(\{H_k\}, \{Y_k\})$. That is, we can combine (i) the fixed point equation of the mean field approximation for $q_k(h_k)$ (*Theorem 1*) and (ii) the injective embedding for $\tilde{\mu}_i = \int_{\mathcal{H}} \phi(H_i) p(h_i|y_i) dh_i$ to come up with parameterized fixed point equation for $\tilde{\mu}_k$ (see Figure 2). As in Dai et al. (2016), we restrict our discussion to the case where there are semi-cliques between two random variables. In this case, the notation we use for $\mathcal{D}_m$ and $p_m$ is $\mathcal{D}_{ij}$ and $p_{ij}$.

*Lemma 1. Structure2vec for random PGM. Given a random PGM on $\mathcal{X} = (\{H_k\}, \{Y_k\})$. As [Dai et al. (2016)], suppose that our problem has a PGM structure with joint distribution proportional to some factorization $\prod_k \phi(h_k, y_k) \prod_{i,j} \phi(h_i, h_j)$. Assume that the presence probabilities $\{p_{ij}\}$ for all pairwise semi-cliques $\mathcal{D}_{ij} \in \mathfrak{C}_{\mathcal{X}}$ are given. Then fixed point equation in Theorem 1 for $p(\{H_k\}|\{y_k\})$ is embedded to generate the fixed point equation $\tilde{\mu}_k = \sigma\left(W_1 y_k + W_2 \sum_{j \neq k} p_{kj} \tilde{\mu}_j\right)$.*

The proof of *Lemma 1* can be found in Appendix A.4.

**Remarks.** Note that inference of $\mathcal{P}$ is in general a difficult task. One implication of *Theorem 1* is that *we transformed a difficult inference task to a simple inference task*: inferring the presence

probability of each semi-clique. (See Appendix A.5 for the algorithm that conducts this task). In addition, *Lemma 1* provides a theoretical justification to ignore the interdependencies among edge presences when embedding a random graph using GNN. When graph edges are not explicitly given or known to be random, the simplest heuristic one can imagine is to separately infer the presence probabilities of all edges and adjust the weights of GNN's message propagation. According to *Lemma 1*, possible interdependencies among edges would not affect the quality of such heuristic's inference.

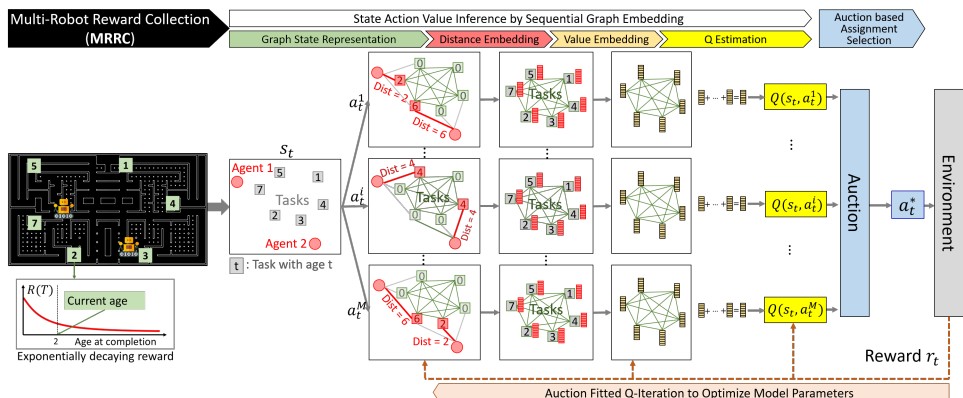

Figure 3: Illustration of overall pipeline of our method

# 4 SCHEDULING MRRC WITH *Random structure2vec*

As illustrated in Appendix A.2, MRRC problems with no randomness induces Bayesian networks with factorization $\prod_k \phi(h_k, y_k) \prod_{i,j} \phi(h_i, h_j)$. Therefore, according to Lemma 1, we are justified to use random structure2vec to design a method to learn solutions to our MRRC problems.

## 4.1 DESIGNING $Q$-FUNCTION ESTIMATOR HAVING ORDER-TRANSFERABILITY

Intuitively, local graph information around node $k$ is embedded into the *structure2vec* output vector $\tilde{\mu}_k$ [Dai et al. (2016)]. Using this intuition, we propose a two-step sequential and hierarchical state-embedding neural network using *random structure2vec* that is designed to achieve what we will later call *order-transferable* Q-function estimation. This allows problem-size transferable Q-learning, i.e., the neural network parameter $\theta$, trained to calculate $Q_\theta^m$ that approximates the Q-function $Q^m$ for an $m$-robot scheduling problem, can be well used to solve $n$-robot scheduling problems ($n \neq m$). For brevity, we assume task completion times are deterministic. For the detailed algorithm with random task completion times, see Appendix A.6. The following procedure is illustrated in Figure 3.

**Step 1. Distance Embedding.** The first *structure2vec* layer embeds information of robot locations around each task $k$, i.e. local graph structure around each task $k$ with respect to robots, to each $\tilde{\mu}_k^1$ (superscript 1 denotes the outcome of first layer). For the input of the first *structure2vec* layer ($\{x_k\}$ in Lemma 1), we only use robot assignment information (if $k$ is an assigned task, we set the value of $x_k$ to task completion time of assignment (a duration); if $k$ is not an assigned task:, we set $x_k = 0$).
**Step 2. Value Embedding.** The second structure2vec layer embeds how much value is likely in the local graph around task $k$ to $\tilde{\mu}_k^2$. Recall that the output vectors of the first *structure2vec* layer, $\{\tilde{\mu}_k^1\}$, carry information about the graph structure of robots locally around each task. For each task $k$, we concatenate task $k$'s age $\eta_t^k$ with $\tilde{\mu}_k^1$ to get $\tilde{\mu}_k'^1$ and use $\{\tilde{\mu}_k'^1\}$ as the input ($\{x_k\}$ in Lemma 1) to the second *structure2vec* layer. Denote the outcome of second structure2vec layer as $\{\tilde{\mu}_k^2\}$.
**Step 3. Computing $Q_\theta(s_{t_k}, a_{t_k})$.** To derive $Q_\theta(s_{t_k}, a_{t_k})$, we aggregate the embedding vectors for all nodes by $\tilde{\mu}^2 = \sum_k \tilde{\mu}_k^2$ to obtain one global vector $\tilde{\mu}^2$ to embed the value affinity of the global graph. We then use a neural network to map $\tilde{\mu}^2$ into $Q_\theta(s_{t_k}, a_{t_k})$.

Let us provide the intuition related to problem-size transferability of Q-learning. Step 1 above, transferability is trivial; the inference problem is a scale-free task *locally around each node*. For Step 2, consider the ratio of robots to tasks. The overall value affinity embedding will be underestimated if this ratio in the training environment is smaller than this ratio in the testing environment;

overestimated overall otherwise. The intuition is that this over/under-estimation does not matter in Q-learning [van Hasselt et al. (2015)] as long as the *order* of Q-function value among actions are the same. That is, as long as the best assignments chosen are the same, i.e., $\text{argmax}_{a_{t_k}} Q^n(s_{t_k}, a_{t_k})$ = $\text{argmax}_{a_{t_k}} Q_\theta^n(s_{t_k}, a_{t_k})$, the magnitude of imprecision $|Q^n(s_{t_k}, a_{t_k}) - Q_\theta^n(s_{t_k}, a_{t_k})|$ does not matter. We call this property *order-transferability* of Q-function estimator with $\theta$.

## 4.2 Order transferability-enabled auction for scalable computation

Learning-based heuristics for solving NP-hard problems have recently received attention due to their fast computation speed for large size NP-hard problems [Dai et al. (2017)]. However, this advantage disappears for Q-learning methods when faced with large action spaces [Lillicrap et al. (2015)]. For multi-robot/machine scheduling problems, the set of all multi-robot assignments at each decision epoch is the action space; it grows exponentially as the number of robots and tasks increases. As such, the computational requirement of the $\text{argmax}_{a_{t_k}} Q(s_{t_k}, a_{t_k})$ operation increases exponentially.

In this section, we demonstrate how order transferability of Q-function estimation enables us to design a polynomial-time algorithm with a provable performance guarantee ($1 - 1/e$ optimality) to substitute for the $\text{argmax}$ operation. We call this algorithm an order transferability-enabled auction-based policy (OTAP) and denote it as $\pi_{Q_\theta}$, where the $Q_\theta$ indicates that the Q-function estimator with current parameter $\theta$ is used during the auction.

### 4.2.1 Order transferability-enabled auction-based policy (OTAP)

We continue to use the notation introduced in section 4.1. Recall that state $s_{t_k} = (\mathcal{G}_{t_k}, \alpha_{t_k})$ where $\mathcal{G}_{t_k} = (\mathcal{R} \cup \mathcal{T}_{t_k}, \mathcal{E}_{t_k}^{RT})$. OTAP finds an assignment $a_t$, the edge set of a maximal bipartite matching in the bipartite graph $\mathcal{G}_{t_k}$, after $N = \max(|\mathcal{R}|, |T_t|)$ iterations of *Bidding* and *Consensus* phases.

**Bidding phase.** In the $n^{th}$ bidding phase, initially *all robots know* $\mathcal{M}_\theta^{(n-1)}$, the ordered set of $n-1$ robot-task edges in $\mathcal{E}_{t_k}^{RT}$ determined by the previous $n-1$ iterations. An unassigned robot $i$ ignores all others unassigned and calculates $Q_\theta^n(s_{t_k}, \mathcal{M}_\theta^{(n-1)} \cup \{\epsilon_{ip}^{RT}\})$ for each unassigned task $p$ as if those $k$ robots (robot $i$ together with all robots assigned tasks in the previous $n-1$ iterations) only exist in the future and will serve all remaining tasks. (Here, $\epsilon_{ip}^{RT} \in \mathcal{E}_{t_k}^{RT}$ is the edge corresponding to assigning robot $i$ to task $j$ at decision epoch $t_k$.) If task $\ell$ has the highest value, robot $i$ bids $\{\epsilon_{i\ell}^{RT}, Q_\theta^n(s_t, \mathcal{M}_\theta^{(n-1)} \cup \{\epsilon_{i\ell}^{RT}\})\}$ to the centralized auctioneer. Since the number of ignored robots varies at each iteration, transferability of Q-function inference is crucial.

**Consensus phase.** At $n^{th}$ consensus phase, the centralized auctioneer finds the bid with the best bid value, say $\{\epsilon_{i^*p^*}^{RT}, Q_\theta^n(s_t, \mathcal{M}_\theta^{(n-1)} \cup \{\epsilon_{i^*p^*}^{RT}\})\}$. (Here $i^*$ and $p^*$ denote the best robot task pair.) Denote $\epsilon_{i^*p^*}^{RT} =: m_\theta^{(n)}$. The centralized auctioneer updates the shared ordered set $\mathcal{M}_\theta^{(n)} = \mathcal{M}_\theta^{(n-1)} \cup m_\theta^{(n)}$.

These two phases iterate until we reach $\mathcal{M}_\theta^{(N)} = \{m_\theta^{(1)}, \ldots, m_\theta^{(N)}\}$. This $\mathcal{M}_\theta^{(N)}$ is chosen as the joint assignment $a_{t_k}^*$ at time step $t_k$. That is, $\pi_{Q_\theta}(s_{t_k}) = a_{t_k}^*$. The computational complexity for computing $\pi_{Q_\theta}$ is $O(|R||T_{t_k}|)$ and is only polynomial (See Appendix A.8.1).

**Provable performance bound of OTAP.**

Let the true Q-functions for OTAP be $\{Q^n\}_{n=1}^N$. Denote the outcome of OTAP with these true Q-functions as $\mathcal{M}^{(N)} = \{m^{(1)}, \ldots, m^{(N)}\}$.

*Lemma 2. If the Q-function approximator has order transferability, then $\mathcal{M}^{(N)} = \mathcal{M}_\theta^{(N)}$.*

For any decision epoch $t_k$, let $\mathcal{M}$ denote a set of robot-task pairs (a subset of $\mathcal{E}_{t_k}^{RT}$). For any robot-task pair $m \in \mathcal{E}_{t_k}^{RT}$, define $\Delta(m \mid \mathcal{M}) := Q^{|\mathcal{M} \cup \{m\}|}(s_{t_k}, \mathcal{M} \cup \{m\}) - Q^{|\mathcal{M}|}(s_{t_k}, \mathcal{M})$ as the the marginal value (under the true Q-functions) of adding robot-task pair $m \in \mathcal{E}_{t_k}^{RT}$. Note, we allow "adding" $m \in \mathcal{M}$ for mathematical convenience in the subsequent proof. In that case, we have $\Delta(m \mid \mathcal{M}) = 0, m \in \mathcal{M}$.

*Theorem 2. Suppose that the Q-function approximation with the parameter value $\theta$ exhibits order transferability. Denote $\mathcal{M}_\theta^{(N)}$ as the result of OTAP using $\{Q_\theta^n\}_{n=1}^N$ and let $\mathcal{M}^* = \text{argmax}_{a_{t_k}}$*

$Q^{|a_{t_k}|}(s_{t_k}, a_{t_k})$. *If $\Delta(m \mid \mathcal{M}) \geq 0, \forall \mathcal{M} \subset \mathcal{E}_{t_k}^{RT}, \forall m \in \mathcal{E}_{t_k}^{RT}$, and the marginal value of adding one robot diminishes as the number of robots increases, i.e., $\Delta(m \mid \mathcal{M}) \leq \Delta(m \mid \mathcal{N}), \forall \mathcal{N} \subset \mathcal{M} \subset \mathcal{E}_{t_k}^{RT}$, $\forall m \in \mathcal{E}_{t_k}^{RT}$, then the result of OTAP is at least better than $1 - 1/e$ of an optimal assignment. That is, $Q_\theta^N(s_{t_k}, \mathcal{M}_\theta^{(N)}) \geq Q^{|\mathcal{M}^*|}(s_{t_k}, \mathcal{M}^*)(1 - 1/e)$.*

For proofs of Lemma 2 and Theorem 2, see Appendix A.7 and A.8.

#### 4.2.2 AUCTION-FITTED Q-ITERATION FRAMEWORK AND EXPLORATION

**Auction-fitted Q-iteration.** We incorporate OTAP into a fitted Q-iteration, i.e., we find $\theta$ that empirically minimizes $E_{\pi_{Q_\theta}, s_{k+1} \sim P'}\left[Q_\theta(s_k, a_k) - \left[r(s_k, a_k) + \gamma Q_\theta(s_{k+1}, \pi_{Q_\theta}(s_{k+1}))\right]\right]$. Please note that this method's rigorous fixed point analysis is the scope of subsequent future research.

**Exploration.** How can we conduct exploration in the auction-fitted Q-iteration framework? Unfortunately, we cannot use an $\epsilon$-greedy method since: (i) an arbitrary random deviation in a joint assignment often induces a catastrophic failure [Maffioli (1986)], and (ii) the joint assignment space, which is complex and combinatorial, is difficult to explore efficiently with such an arbitrary random exploration policy. In learning the parameters $\theta$ for $Q_\theta(s_k, a_k)$, we use the exploration strategy that perturbs the parameters $\theta$ randomly to actively explore the joint assignment space with TAP. While this method was originally developed for policy-gradient based methods [Plappert et al. (2017)], exploration in parameter space is useful in our auction-fitted Q-iteration since it generates a reasonable combination of assignments.

## 5 EXPERIMENTS AND RESULTS

We focus on DTDS MRRC problem in the main paper and now elaborate upon our reasoning. As discussed in section 2, the formulation of MRRC problem assumes that task completion times are given as prior knowledge. Recall that in stochastic environment, task completion times are given as random variables or sets of samples. In a simulation experiment standpoint, one must generate such a dataset of task completion times before she can discuss algorithms to solve MRRC problems. *However, it is extremely difficult* to generate a reasonable distribution of task completion times under continuous state continuous time (CTCS) environment [Bertsekas (2014); Omidshafiei et al. (2017)]. For example, finding an optimal control for stochastic routing problems under a CTCS environment is in general intractable unless you discretize the space and time so that you transform CTCS environment to get an approximate DTDS environment [Kushner & Dupuis (2013)].

Despite this difficulty, stochastic environment experiment is important since one of the main benefits of learning-based heuristics is its capability to tractably solve stochastic scheduling problems [Rossi et al. (2018)]. Therefore, in this paper, we focus on experiments under DTDS environment and target to show that our algorithm's performance for deterministic environments extends to stochastic environments. Since there is no standard dataset for MRRC problems, we deliberately created a grid-world environment that generates nontrivial task completion time distributions with minimizing the selection bias. The idea we took was to use a complex maze (see Figure 3) generator of Neller et al. (2010) (code provided in Appendix 10) and compare it with the baselines. (For CTCS environment experiment under deterministic environment, refer IPMS experiments (Appendix A.1)). We avoided over-fitting by randomly generating a new maze for every training and testing experiment with initial task/robot locations also randomly chosen, only fixing the problem size while doing that.

To generate the task completion times, Dijkstra's algorithm and dynamic programming were used for deterministic and stochastic environments, respectively. To minimize artificiality, the simplest MRRC problem is considered as follows. In the deterministic environment, robots always succeed in their movement. In the stochastic environment, a robot makes its intended move with a certain probability. (Cells with a dot: success with 55%, every other direction with 15% each. Cells without a dot: 70% and 10%, respectively.) A task is considered served when a robot reaches it. We consider two reward rules: linearly decaying rewards $f(age) = \max\{200 - age, 0\}$ and nonlinearly decaying rewards $f(age) = \lambda^{age}$ with $\lambda = 0.99$, where $age$ is the task age when served. The initial age of tasks are uniformly distributed in the interval $[0, 100]$. Throughout, the performance measure used is $\rho =$ (%rewards collected by the proposed method/reward collected by the baseline). The baselines are:
- *%Optimal*: Gurobi was used for problems with the deterministic environment and linear rewards. Gurobi Optimization (2019) was allowed a 60-min time limit to search for an optimal solution.

- *Ekici et al*: For deterministic environments with linear rewards, an up-to-date, fast heuristic for MRRC (Ekici & Retharekar (2013)) was used (it claims 93% optimality for 50 tasks and 4 robots).
- *Sequential Greedy Algorithm (SGA)*: To our knowledge, there is no literature addressing MRRC with stochastic environments or exponential rewards. Instead, we construct an indirect baseline using a general-purpose multi-robot task allocation algorithm called SGA (Han-Lim Choi et al. (2009)). We will provide our performance divided by SGA performance as %SGA. We will see that the %SGA in the deterministic linear-reward case is maintained for other cases.

**Performance test.** We tested the performance under four environments: deterministic/linear rewards, deterministic/nonlinear rewards, stochastic/linear rewards, stochastic/nonlinear rewards. See Table 1. For linear/deterministic rewards, our method achieves near-optimality with 3% fewer rewards than *optimal* on average. The standard deviation for $\rho$ is provided in parentheses. For others, we see that the %SGA ratio for linear/deterministic is well maintained in stochastic or nonlinear environments. Due to dynamic programming computation complexity of dataset generation, we only consider 8 robots/50 tasks at maximum. Larger size problems were considered in IPMS experiments.

Table 1: Performance test (50 trials of training for each cases)

| Reward | Environment | Baseline | Testing size : Robot (R) / Task (T) | | | | | | |
|---|---|---|---|---|---|---|---|---|---|
| | | | 2R/20T | 3R/20T | 3R/30T | 5R/30T | 5R/40T | 8R/40T | 8R/50T |
| Linear | Deterministic | %Optimal (Gurobi 60min) | 98.31 | 97.50 | 97.80 | 95.35 | 96.99 | 96.11 | 96.85 |
| | | %Ekisi et al. | 99.86 | 97.50 | 118.33 | 110.42 | 105.14 | 104.63 | 120.16 |
| | | %SGA | 137.3 | 120.6 | 129.7 | 110.4 | 123.0 | 119.9 | 119.8 |
| | Stochastic | %SGA | 130.9 | 115.7 | 122.8 | 115.6 | 122.3 | 113.3 | 115.9 |
| Nonlinear | Deterministic | %SGA | 111.5 | 118.1 | 118.0 | 110.9 | 118.7 | 111.2 | 112.6 |
| | Stochastic | %SGA | 110.8 | 117.4 | 119.7 | 111.9 | 120.0 | 110.4 | 112.4 |

**Transferability test.** Table 2 shows comprehensive transferability test results. The rows indicate training conditions, while the columns indicate testing conditions. The results in the diagonal cells in red (cells with the same training size and testing size) serve as baselines (direct testing). The results in the off-diagonal show the results for the transferability testing, and demonstrate how the algorithms trained with different problem size perform well on test problems. We can see that lower-direction transfer tests (trained with larger size problems and tested with smaller size problems) show only a small loss in performance. For upper-direction transfer tests (trained with smaller size problems and tested with larger size problems), the performance loss was up 4 percent.

**Scalability analysis.** For scalability considerations, including computational analysis of OTAP and training data complexity, see Appendix A.8.1.

Table 2: Transferability test (50 trials of training for each cases, linear & deterministic env.)

| Training size (Robot(R)/Task(T)) | Testing size : Robot (R) / Task (T) | | | | | | |
|---|---|---|---|---|---|---|---|
| | 2R/20T | 3R/20T | 3R/30T | 5R/30T | 5R/40T | 8R/40T | 8R/50T |
| 2R/20T | 98.31 | 93.61 | 97.31 | 92.16 | 92.83 | 90.94 | 93.44 |
| 3R/20T | 95.98 | 97.50 | 96.11 | 93.64 | 91.75 | 91.60 | 92.77 |
| 3R/30T | 94.16 | 96.17 | 97.80 | 94.79 | 93.19 | 93.14 | 93.28 |
| 5R/30T | 97.83 | 94.89 | 96.43 | 95.35 | 93.28 | 92.63 | 92.40 |
| 5R/40T | 97.39 | 94.69 | 95.22 | 93.15 | 96.99 | 94.96 | 93.65 |
| 8R/40T | 95.44 | 94.43 | 93.48 | 93.93 | 96.41 | 96.11 | 95.24 |
| 8R/50T | 95.69 | 96.68 | 97.35 | 94.02 | 94.50 | 94.86 | 96.85 |

## 6 Concluding Remarks

We developed a theory of random PGM-based mean-field inference method and provided a theoretical justification for a simple modification of popular GNN methods to embed a random graph. This theory was motivated from addressing the challenge of developing a near-optimal learning-based algorithm for solving NP-hard multi-robot/machine scheduling problems. While precise inference of $Q$-function is required to address this challenge, the two-layer random structure2vec embedding procedure we suggested has shown an empirical success. We further address inscalability problem of $Q$-learning methods for multi-robot/machine scheduling problem by suggesting a polynomial-time assignment algorithm with a provable performance guarantee.

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

# A  APPENDIX

## A.1  IDENTICAL PARALLEL MACHINE SCHEDULING PROBLEM (IPMS) WITH MAKESPAN MINIMIZATION OBJECTIVE

### A.1.1  FORMULATION

IPMS is a problem defined in continuous state/continuous time space. Once service of a task $i$ begins, it requires a deterministic duration of time $\tau_i$ for a machine to complete - we call this the processing time. Machines are all identical, which means processing time of each tasks among machines are all the same. Processing times of each tasks are all different. Before a machine can start processing a task, it is required to first setup for the task. In this paper, we discuss IPMS with 'sequence-dependent setup times'. In this case, a machine must conduct a setup prior to serving each task. The duration of this setup depends on the current task $i$ and the task $j$ that was previously served on that machine - we call this the setup time. The completion time for each task is thus the sum of the setup time and processing time. Under this setting, we solve the IPMS problem for make-span minimization as discussed in [Kurz et al. (2001)]. That is, we seek to minimize the total time spent from the start time to the completion of the last task. IPMS problem's sequential decision making problem formulation resembles that of MRRC with continuous-time and continuous-space. That is, every time there is a finished task, we make assignment decision for a free machine. We call this times as 'decision epochs' and express them as an ordered set $(t_1, t_2, \ldots, t_k, \ldots)$. Abusing this notation slightly, we use $(\cdot)_{t_k} = (\cdot)_k$. This problem can be cast as a Markov Decision Problem (MDP) whose state, action, and reward are defined as follows:

**State.** The state $s_{t_k}$ at time $t_k$ is represented as $\left(\mathcal{G}_{t_k}, \mathcal{W}_{t_k}^{RT}, t_k\right)$. $\mathcal{G}_t$ is a directed bipartite graph $(\mathcal{R} \cup \mathcal{T}_{t_k}, \mathcal{E}_{t_k}^{RT})$ where $\mathcal{R}$ is the set of all machines, $\mathcal{T}_{t_k}$ is the set of all remaining unserved tasks at time step $t_k$. The set $\mathcal{E}_{t_k}^{RT}$ consists of all directed edges from machines to unserved tasks at time $t_k$. To each edge is associated a weight equal to the task completion time. Let $\mathcal{W}_{t_k}^{RT}$ denote the set of all such weights for all edges at $t_k$ (either constants or random variables and restricted to multiples of $\Delta$ in the the DTDS system). For example, $\epsilon_{i,p}^{RT} \in \mathcal{E}_{t_k}^{RT}$ is an edge indicating machine $i$ is assigned to serve task $p$. To this edge a random variable denoting the task completion time (a duration) is assigned. Each task is given an initial age which increases linearly with time (a multiple of $\Delta$ for DTDS). Let $\alpha_{t_k} = \{\eta_{t_k}^p \in \mathbb{R} | p \in \mathcal{T}_{t_k}\}$ denote the set of ages where $\eta_{t_k}^p$ indicates the age of task $p$ at time-step $t_k$. We denote the set of possible states as $\mathcal{S}$.

**Action.** Defined the same as MRRC with continuous state/time space.

**Reward.** Let's denote the time between decision epoch $k$ and decision epoch $k+1$ as $T_k = t_k - t_{k-1}$. One can easily see that $T_k$ is completely determined by $s_k$, $a_k$ and $s_{k+1}$. Therefore, we can denote the reward we get with $s_k$, $a_k$ and $s_{k+1}$ as $T(s_k, a_k, s_{k+1})$.

**Transition probabilities.** The transition probability $P'$ is defined the same as MRRC problem.

**Objective.** We can now define an assignment policy $\phi$ as a function that maps a state $s_k$ to action $a_k$. Given $s_0$ initial state, an IPMS problem with makespan minimization objective can be expressed as a problem of finding an optimal assignment policy $\phi^*$ such that

$$\phi^* = \operatorname*{argmin}_{\phi} \mathbb{E}_{\pi, P'}\left[\sum_{k=0}^{\infty} T(s_k, a_k, s_{k+1}) | s_0\right].$$

Table 3: IPMS test results for makespan minimization with deterministic task completion time (our algorithm / best Google OR tool result)

| Makespan minimization | | # Machines | | | |
|---|---|---|---|---|---|
| | | 3 | 5 | 7 | 10 |
| **# Tasks** | 50 | 106.7% | 117.0% | 119.8% | 116.7% |
| | 75 | 105.2% | 109.6% | 113.9% | 111.3% |
| | 100 | 100.7% | 111.0% | 109.1% | 109.0% |

### A.1.2 EXPERIMENTS

For IPMS, we test it with continuous time, continuous state environment. While there have been many learning-based methods proposed for (single) robot scheduling problems, to the best our knowledge our method is the first learning method to claim scalable performance among machine-scheduling problems. Hence, in this case, we focus on showing comparable performance for large problems, instead of attempting to show the superiority of our method compared with heuristics specifically designed for IPMS (actually no heuristic was specifically designed to solve our exact problem (makespan minimization, sequence-dependent setup with no restriction on setup times))

For each task, processing times is determined using uniform [16, 64]. For every (task $i$, task $j$) ordered pair, a unique setup time is determined using uniform [0, 32]. As illustrated in Appendix A.1, we want to minimize make-span. As a benchmark for IPMS, we use Google OR-Tools library Google (2012). This library provides metaheuristics such as Greedy Descent, Guided Local Search, Simulated Annealing, Tabu Search. We compare our algorithm's result with the heuristic with the best result for each experiment. We consider cases with $3, 5, 7, 10$ machines and $50, 75, 100$ jobs.

The results are provided in Appendix Table 3. Makespan obtained by our method divided by the makespan obtained in the baseline is provided. Although our method has limitations in problems with a small number of tasks, it shows comparable performance to a large number of tasks and shows its value as the first learning-based machine scheduling method that achieves scalable performance.

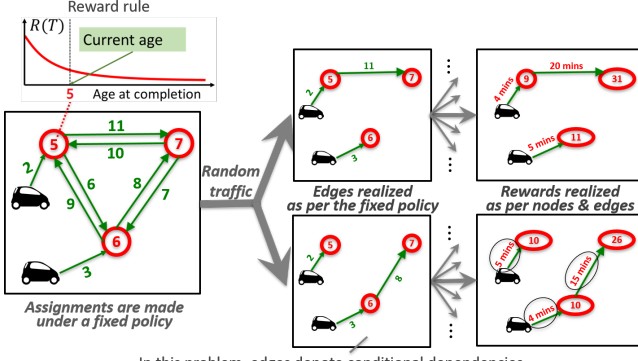

Figure 4: Representing MRRC as a random Bayesian Network

### A.2 BAYESIAN NETWORK REPRESENTATION

Here we illustrate that robot scheduling problem randomly induces a random Bayesian Network from state $s_t$. See figure 4. Given starting state $s_t$ and action $a_t$, a person can repeat a random experiment of "sequential decision making using policy $\phi$". In this random experiment, we can define events 'How robots serve all remaining tasks in which sequence'. We call such an event a 'scenario'. For example, suppose that at time-step $t$ we are given robots $\{A, B\}$, tasks $\{1, 2, 3, 4, 5\}$, and policy $\phi$. One possible scenario $S^*$ can be {robot A serves task $3 \rightarrow 1 \rightarrow 2$ and robot B serves task $5 \rightarrow 4$}. Define random variable $\{\{H_j\}$ a task characteristic, e.g. 'The time when task $k$ is serviced'. The question is, 'Given a scenario $S^*$, what is the relationship among random variables $\{H_k\}$' $\{y_k\}$ (inputs in section 4.1)? Recall that in our sequential decision making formulation we are given all the 'task completion time' information in the $s_t$ description. Note that, task completion time is only dependent

on the previous task and assigned task. In our example above, under scenario $S^*$ 'when task 2 is served' is only dependent on 'when task 1 is served'. That is, $P(H_2|H_1, H_3, S^*) = P(H_2|H_1, S^*)$. This relationship is called 'conditional independence'. Given a scenario $S^*$, every relationship among $\{H_i|S^*\}$ can be expressed using this kind of relationship among random variables. A graph with this special relationship is called 'Bayesian Network' [Koller & Friedman (2009)], a probabilistic graphical model. Therefore, under a fixed scenario $S^*$, this problem's joint distribution can be assumed to be factored as PGM structure $\prod_k \phi(h_k|y_k) \prod_{i,j} \phi(h_i|h_j)$ where $y_k$ is the inputs considered in section 4.1 and $H_i$ denoting the time task $i$ is served.

### A.3 PROOF OF THEOREM 1.

We first define necessary definitions for our proof. Given a random PGM $\{\mathcal{G}_\mathcal{X}, \mathcal{P}\}$, a PGM is chosen among $\mathcal{G}_\mathcal{X}$, the set of all possible PGMs on $\mathcal{X}$. The set of semi-cliques is denoted as $\mathfrak{C}_\mathcal{X}$. As discussed in the main text, if we are given $\mathcal{P}$ then we can easily calculate the presence probability $p_m$ of semi-clique $\mathcal{D}_m$ as $p_m = \sum_{G \in \mathcal{G}_\mathcal{X}} \mathcal{P}(G) 1_{\mathcal{D}_m \in G}$.

For each semi-clique $\mathcal{D}^i$ in $\mathfrak{C}_\mathcal{X}$, define a binary random variable $V^i \colon \mathcal{F} \mapsto \{0, 1\}$ with value 0 for the factorization that does not include semi-clique $\mathcal{D}^i$ and value 1 for the factorization that include semi-clique $\mathcal{D}^i$. Let $V$ be a random vector $V = \left(V^1, V^2, \ldots, V^{|\mathfrak{C}_\mathcal{X}|}\right)$. Then we can express $P(X_1, \ldots, X_n|V) \propto \prod_{i=1}^{|\mathfrak{C}_\mathcal{X}|} \left[\phi^i\left(\mathcal{D}^i\right)\right]^{V^i}$. We denote $\left[\phi^i\left(\mathcal{D}^i\right)\right]^{V^i}$ as $\psi(\mathcal{D}^i)$.

Now we prove Theorem 1.

In mean-field inference, we want to find a distribution $Q(X_1, \ldots, X_n) = \prod_{i=1}^n Q_i(X_i)$ such that the cross-entropy between it and a target distribution is minimized. Following the notation in Koller & Friedman (2009), the mean field inference problem can written as the following optimization problem.

$$\min_Q \quad \mathbb{D}\left(\prod_i Q_i \,| P(X_1, \ldots, X_n|V))\right)$$
$$\text{s.t.} \quad \sum_{x_i} Q_i(x_i) = 1 \quad \forall i$$

Here $\mathbb{D}\left(\prod_i Q_i \mid P(X_1, \ldots, X_n|V)\right)$ can be expressed as $\mathbb{D}\left(\prod_i Q_i \mid P(X_1, \ldots, X_n|V)\right) = \mathbb{E}_Q\left[\ln\left(\prod_i Q_i\right)\right] - \mathbb{E}_Q\left[\ln\left(P(X_1, \ldots, X_n|V)\right)\right]$.

Note that

$$
\mathbb{E}_Q\left[\ln\left(P\left(X_1,\ldots,X_n|V\right)\right)\right] = \mathbb{E}_Q\left[\ln\left(\frac{1}{z}\Pi_{i=1}^{|\mathfrak{C}_\mathcal{X}|}\psi^i\left(\mathcal{D}^i,V\right)\right)\right]
$$

$$
= \mathbb{E}_Q\left[\ln\left(\frac{1}{z}\prod_{i=1}^{|\mathfrak{C}_\mathcal{X}|}\psi^i\left(\mathcal{D}^i,V\right)\right)\right]
$$

$$
= \mathbb{E}_Q\left[\sum_{i=1}^{|\mathfrak{C}_\mathcal{X}|}V^i\ln\left(\phi^i\left(\mathcal{D}^i\right)\right)\right] - \mathbb{E}_Q[\ln(Z)]
$$

$$
= \sum_{i=1}^{|\mathfrak{C}_\mathcal{X}|}\mathbb{E}_Q\left[V^i\ln\left(\phi^i\left(\mathcal{D}^i\right)\right)\right] - \mathbb{E}_Q[\ln(Z)]
$$

$$
= \sum_{i=1}^{|\mathfrak{C}_\mathcal{X}|}\mathbb{E}_{V^i}\left[\mathbb{E}_Q\left[V^i\ln\left(\phi^i\left(\mathcal{D}^i\right)\right)|V^i\right]\right] - \mathbb{E}_Q[\ln(Z)]
$$

$$
= \sum_{i=1}^{|\mathfrak{C}_\mathcal{X}|}P\left(V^i=1\right)\left[\mathbb{E}_Q\left[\ln\left(\phi^i\left(\mathcal{D}^i\right)\right)\right]\right] - \mathbb{E}_Q[\ln(Z)]
$$

$$
= \sum_{i=1}^{|\mathfrak{C}_\mathcal{X}|}p_i\left[\mathbb{E}_Q\left[\ln\left(\phi^i\left(\mathcal{D}^i\right)\right)\right]\right] - \mathbb{E}_Q[\ln(Z)].
$$

Hence, the above optimization problem can be written as

$$
\max_Q \quad \mathbb{E}_Q\left[\sum_{i=1}^{|\mathfrak{C}_\mathcal{X}|}p_i\ln\left(\phi^i\left(\mathcal{D}^i\right)\right)\right] + \mathbb{E}_Q\sum_{i=1}^{n}\left(\ln Q_i\right) \tag{1}
$$

$$
\text{s.t.} \quad \sum_{x_i}Q_i\left(x_i\right) = 1 \quad \forall i
$$

In Koller & Friedman (2009), the fixed point equation is derived by solving an analogous equation to (1) without the presence of the $p_i$. Theorem 1 follows by proceeding as in Koller & Friedman (2009) with straightforward accounting for $p_i$.

### A.4 PROOF OF LEMMA 1.

Since we assume semi-cliques are only between two random variables, we can denote $\mathfrak{C}_\mathcal{X} = \{\mathcal{D}^{ij}\}$ and presence probabilities as $\{p_{ij}\}$ where $i,j$ are node indexes. Denote the set of nodes as $\mathcal{V}$.

From here, we follow the approach of Dai et al. (2016) and assume that the joint distribution of random variables can be written as

$$
p\left(\{H_k\},\{X_k\}\right) \propto \prod_{k\in\mathcal{V}}\psi^i\left(H_k|X_k\right)\prod_{k,i\in\mathcal{V}}\psi^i\left(H_k|H_i\right).
$$

Expanding the fixed-point equation for the mean field inference from Theorem 1, we obtain:

$$
Q_k\left(h_k\right) =
$$

$$
\frac{1}{Z_k}\exp\left\{\sum_{\psi^i:H_k\in\mathcal{D}^i}\mathbb{E}_{(\mathcal{D}^i-\{H_k\})\sim Q}\left[\ln\psi^i\left(H_k=h_k|\mathcal{D}^i\right)\right]\right\}
$$

$$
= \frac{1}{Z_k}\exp\{ln\phi\left(H_k=h_k|x_k\right) +
$$

$$
\sum_{i\in\mathcal{V}}\int_\mathcal{H}p_{ki}Q_i\left(h_i\right)\ln\phi\left(H_k=h_k|H_i\right)dh_i\}.
$$

This fixed-point equation for $Q_k(h_k)$ is a function of $\{Q_j(h_j)\}_{j \neq k}$ such that

$$Q_k(h_k) = f\left(h_k, x_k, \{p_{kj}Q_j(h_j)\}_{j \neq k}\right).$$

As in Dai et al. (2016), this equation can be expressed as a Hilbert space embedding of the form

$$\tilde{\mu}_k = \tilde{\mathcal{T}} \circ \left(x_k, \{p_{kj}\tilde{\mu}_j\}_{j \neq i}\right),$$

where $\tilde{\mu}_k$ indicates a vector that encodes $Q_k(h_k)$. In this paper, we use the nonlinear mapping $\tilde{\mathcal{T}}$ (based on a neural network form ) suggested in Dai et al. (2016):

$$\tilde{\mu}_k = \sigma\left(W_1 x_k + W_2 \sum_{j \neq k} p_{kj}\tilde{\mu}_j\right)$$

### A.5 SIMPLE PRESENCE PROBABILITY INFERENCE METHOD USED FOR MRRC

Denote ages of task $i, j$ as $age_i, age_j$. Note that if we generate M samples of $\epsilon_{ij}$ as $\{e_{ij}^k\}_{k=1}^M$, then $\frac{1}{M}\sum_{k=1}^M f(e_{ij}^k, age_i, age_j)$ is an unbiased and consistent estimator of $E[f(\epsilon_{ij}, age_i, age_j)]$. The corresponding neural network-based inference is as follows: for each sample $k$, for each task $i$ and task $j$, we form a vector of $u_{ij}^k = (e_{ij}^k, age_i, age_j)$ and compute $g_{ij} = \sum_{k=1}^M \frac{1}{M}W_1(relu(W_2 u_{ij}^k)$. We obtain $\{p_{ij}\}$ from $\{g_{ij}\}$ using softmax.

The pseducode implementation is as follows: In lines 1 and 2, the likelihood of the existence of a directed edge from each node $m$ to node $n$ is computed by calculating $W_1\left(relu\left(W_2 u_{mn}^k\right)\right)$ and averaging over the $M$ samples. In lines 3 and 4, we use the soft-max function to obtain $p_{m,n}$.

1  For $m, n \in \mathcal{V}$ do
2  $\quad g_{mn} = \frac{1}{M}\sum_{k=1}^M W_1\left(relu\left(W_2 u_{mn}^k\right)\right)$
3  For $m, n \in \mathcal{V}$ do
4  $\quad p_{m,n} = \frac{e^{g_{mn}/\tau}}{\sum_{j \in v} e^{g_{mn}/\tau}}.$

### A.6 COMPLETE ALGORITHM OF SECTION 4.1 WITH TASK COMPLETION TIME AS A RANDOM VARIABLE

We combine random sampling and inference procedure suggested in section and Figure 3. Denote the set of task with a robot assigned to it as $\mathcal{T}^A$. Denote a task in $\mathcal{T}^A$ as $t_i$ and the robot assigned to $t_i$ as $r_{t_i}$. The corresponding edge in $\mathcal{E}^{RT}$ for this assignment is $\epsilon_{r_{t_i}t_i}$. The key idea is to use samples of $\epsilon_{r_{t_i}t_i}$ to generate $N$ number of sampled $Q(s,a)$ value and average them to get the estimate of $E(Q(s,a))$. First, for $l = 1\ldots N$ we conduct the following procedure. For each task $t_i$ in $\mathcal{T}^A$, we sample one data $e_{r_{t_i}t_i}^l$. Using those samples and $\{p_{ij}\}$, we follow the whole procedure illustrated in section 4.1 to get $Q(s,a)^l$. Second, we get the average of $\{Q(s,a)^l\}_{l=1}^{l=N}$ to get the estimate of $E(Q(s,a))$, $\frac{1}{N}\sum_{l=1}^{l=N} Q(s,a)^l$.

The complete algorithm of section 4.1 with task completion time as a random variable is given as below.

1  $age_i = $ age of node $i$
2  *The set of nodes for assigned tasks* $\equiv \mathcal{T}_A$
3  *Initialize* $\{\tilde{\mu}_i^{(0)}\}, \{\gamma_i^{(0)}\}$
4  for $l = 1$ to $N$:
5  $\quad$ for $t_i \in \mathcal{T}$:
5  $\quad\quad$ if $t_i \in \mathcal{T}^A$ do:
6  $\quad\quad\quad$ sample $e_{r_{t_i}t_i}^l$ from $\epsilon_{r_{t_i}t_i}$
7  $\quad\quad\quad$ $x_i = e_{r_{t_i}t_i}^l$
9  $\quad\quad$ else: $x_i = 0$
10 $\quad$ for $t = 1$ to $T_1$ do
11 $\quad\quad$ for $i \in \mathcal{V}$ do

12        $l_i = \sum_{j \in \mathcal{V}} p_{ji} \tilde{\mu}_j^{(t-1)}$

13        $\tilde{\mu}_i^{(t)} = relu\left(W_3 l_i + W_4 x_i\right)$

14       $\widetilde{\mu}_l = $ Concatenate $\left(\tilde{\mu}_i^{(T_1)}, age_i\right)$

15      for $t = 1$ to $T_2$ do

16        for $i \in \mathcal{V}$ do

17         $l_i = \sum_{j \in \mathcal{V}} p_{ji} \gamma_j^{(t-1)}$

18         $\gamma_j^{(t)} = relu\left(W_5 l_i + W_6 \tilde{\mu}_i\right)$

19     $Q_l = W_7 \sum_{i \in \mathcal{V}} \gamma_i^{(T)}$

20   $Q_{avg} = \frac{1}{N} \sum_{l=1}^{N} Q_l$

## A.7 Proof of Lemma 2

**Statement:** *Denote result of OTAP using true Q-functions $\{Q^{(n)}\}$ as $\mathcal{M}^{(N)} = \{m^{(1)} \ldots m^{(N)}\}$. If Q-function approximation method has order transferability, then $\mathcal{M}^{(N)} = \mathcal{M}_\theta^{(N)}$ holds.*

**Proof.** Recall that we say Q-function approximation method has order transferability if $\text{argmax}_{a_{t_k}} Q^n(s_{t_k}, a_{t_k}) = \text{argmax}_{a_{t_k}} Q_\theta^n(s_{t_k}, a_{t_k})$. We prove by induction.

Base case: For $n = 0$, $\mathcal{M}^{(0)} = \phi = \mathcal{M}_\theta^{(0)}$.

For $n > 0$, suppose that $\mathcal{M}^{(n)} = \mathcal{M}_\theta^{(n)}$ holds, i.e. $m^{(j)} = m_\theta^{(j)}$ for $1 \leq j \leq n$. Then according to $n + 1^{th}$ step OTEP operation,

$m^{(n+1)} = \text{argmax}_m Q^{n+1}\left(s_{t_k}, \mathcal{M}^{(n)} \cup \{m\}\right)$

$= \text{argmax}_m Q_\theta^{n+1}\left(s_{t_k}, \mathcal{M}^{(n)} \cup \{m\}\right)$ ($\because$ Order transferability assumption)

$= \text{argmax}_m Q_\theta^{n+1}\left(s_{t_k}, \mathcal{M}_\theta^{(n)} \cup \{m\}\right)$ ($\because$ induction argument)

$= m_\theta^{(n+1)}$.

Therefore, $\mathcal{M}^{(n+1)} = \mathcal{M}^{(n)} \cup \{m^{(n+1)}\} = \mathcal{M}_\theta^{(n)} \cup \{m_\theta^{(n+1)}\} = \mathcal{M}_\theta^{(n+1)}$.

## A.8 Proof of Theorem 2

**Statement:** *Denote $N = \max\left(|\mathcal{R}|, |T_t|\right)$.*

*Suppose that Q-function approximation method has order transferability. Denote $\mathcal{M}_\theta^{(N)}$ as the result of OTAP using $\{Q_\theta^n\}$ and $\mathcal{M}^*$ as $\text{argmax}_{a_{t_k}} Q(s_{t_k}, a_{t_k})$. If 1) the marginal value of adding one robot is positive, i.e. $Q^{|\mathcal{M}|+1}(s_{t_k}, \mathcal{M} \cup \{m\}) - Q^{|\mathcal{M}|}(s_{t_k}, \mathcal{M}) \geq 0$ for all $\mathcal{M} \subset \mathcal{E}_t^{RT}$ and 2) the marginal value of adding one robot diminishes as the robot number increases, i.e., $Q^{|\mathcal{M}|+1}(s_{t_k}, \mathcal{M} \cup \{m\}) - Q^{|\mathcal{M}|}(s_{t_k}, \mathcal{M}) \leq Q^{|\mathcal{N}|+1}(s_{t_k}, \mathcal{N} \cup \{m\}) - Q^{|\mathcal{N}|}(s_{t_k}, \mathcal{N})$ for $\mathcal{N} \subset \mathcal{M} \subset \mathcal{E}_t^{RT}$, for all $m \in \mathcal{E}_t^{RT}$, then the result of OTAP is at least better than $1 - 1/e$ of optimal assignment, i.e., $Q_\theta^N(s_{t_k}, \mathcal{M}_\theta^{(N)}) \geq Q^{|\mathcal{M}^*|}(s_{t_k}, \mathcal{M}^*)(1 - 1/e)$.*

**Proof.** From the assumption 1) that the marginal value of adding one robot is nonnegative, without loss of generality, we can consider $\mathcal{M}^*$ with $|\mathcal{M}^*| = N$ in the further proof procedure. Denote $\mathcal{M}^* = \{m^{(1)*}, m^{(2)*}, \ldots, m^{(n)*}\}$ and denote $\mathcal{M}_\theta^{(N)} = \{m_\theta^{(1)}, m_\theta^{(2)}, \ldots, m_\theta^{(N)}\}$.

For notation simplicity, define $\Delta(m \mid \mathcal{M}) =: Q^{|\mathcal{M} \cup \{m\}|}(s_t, \mathcal{M} \cup \{m\}) - Q^{|\mathcal{M}|}(s_t, \mathcal{M})$.

Then the optimal value $OPT = Q^N(s_{t_k}, \mathcal{M}^*) \leq Q^{|\mathcal{M}_\theta^{(n)} \cup \mathcal{M}^*|}(s_{t_k}, \mathcal{M}_\theta^{(n)} \cup \mathcal{M}^*)$

$= Q^n(s_{t_k}, \mathcal{M}_\theta^{(n)}) + \sum_{j=1}^{N} \Delta(m^{(j)*} \mid \mathcal{M}_\theta^{(n)} \cup \{m^{(1)*}, \cdots, m^{(j-1)*}\})$

$\leq Q^n(s_{t_k}, \mathcal{M}_\theta^{(n)}) + \sum_{j=1}^{N} \Delta(m^{(j)*} \mid \mathcal{M}_\theta^{(n)})$ ($\because$ condition 2 - decreasing marginal value condition)

$\leq Q^n(s_{t_k}, \mathcal{M}_\theta^{(n)}) + \sum_{j=1}^{N} \Delta(m_\theta^{(n+1)} \mid \mathcal{M}_\theta^{(n)})$

($\because$ OTAP chooses $m_\theta^{(n+1)} = \text{argmax}_m Q_\theta^{n+1}\left(s_t, \mathcal{M}_\theta^{(n)} \cup \{m\}\right)$ and

$\text{argmax}_m Q_\theta^{n+1}\left(s_t, \mathcal{M}_\theta^{(n)} \cup \{m\}\right) = \text{argmax}_m Q^n\left(s_t, \mathcal{M}_\theta^{(n)} \cup \{m\}\right)$ from *Lemma 2*)

$= Q^n(s_{t_k}, \mathcal{M}_\theta^{(n)}) + N\Delta(m_\theta^{(n+1)} \mid \mathcal{M}_\theta^{(n)})$.

Therefore, $\Delta(m_\theta^{(n+1)} \mid \mathcal{M}_\theta^{((n))}) \geq \frac{1}{N}(OPT - Q^n(s_{t_k}, \mathcal{M}_\theta^{(n)})$.

Note that $OPT - Q^n(s_{t_k}, \mathcal{M}_\theta^{(n)})$ denotes current iteration $(= n^{th})$ outcome $\mathcal{M}_\theta^{(n)}$'s size of sub-optimality compared to $OPT$. Denote $OPT - Q^n(s_{t_k}, \mathcal{M}_\theta^{(n)}) =: \beta_n$. Then since $Q^0(s_{t_k}, \phi) = 0$, $\beta_0 = OPT$. Therefore, we have $\Delta(m_\theta^{(n+1)} \mid \mathcal{M}_\theta^{((n))}) \geq \frac{1}{N}\beta_n$.

Also, note that $\Delta(m_\theta^{(n+1)} \mid \mathcal{M}_\theta^{(n)}) = Q^{n+1}(s_t, \mathcal{M}_\theta^{(n)} \cup \{m_\theta^{(n+1)}\}) - Q^n(s_t, \mathcal{M}_\theta^{(n)})$
$= Q^{n+1}(s_t, \mathcal{M}_\theta^{(n+1)}) - Q^n(s_t, \mathcal{M}_\theta^{(n)}) = (OPT - Q^n(s_t, \mathcal{M}_\theta^{(n)})) - (OPT - Q^{n+1}(s_t, \mathcal{M}_\theta^{(n+1)}))$
$= \beta_n - \beta_{n+1}$.

Therefore, $\beta_n - \beta_{n+1} \geq \frac{1}{N}\beta_n$, i.e., $\beta_{n+1} \leq \beta_n \left(1 - \frac{1}{N}\right)$.

This implies $OPT - Q^N(s_{t_k}, \mathcal{M}_\theta^{(N)}) = \beta_N \leq \beta_0(1 - \frac{1}{N})^N = OPT(1 - \frac{1}{N})^N$ and thus we get $Q^N(s_{t_k}, \mathcal{M}_\theta^{(N)}) = OPT(1 - (1 - \frac{1}{N})^N) \sim OPT(1 - \frac{1}{e})$ as $N \to \infty$.

### A.8.1 SCALABILITY ANALYSIS

**Computational complexity**. MRRC can be formulated as a semi-MDP (SMDP) based multi-robot planning problem (e.g., Omidshafiei et al. (2017)). This problem's complexity with $R$ robots and $T$ tasks and maximum H time horizon is $O((R!/T!(R - T)!)^H)$. For example, Omidshafiei et al. (2017) state that a problem with only 13 task completion times ('TMA nodes' in their language) possessed a policy space with cardinality $5.622 * 10^{17}$. In our proposed method, this complexity is addressed by a combination of two complexities: computational complexity and training complexity. For computational complexity of joint assignment decision at each timestep, it is $O(|R||T|^3) = O((1) \times (2) \times (3) \times (4) + (5))$ where $(1) - (5)$ are as follows.

(1) # of Q-function computation required in one time-step = $O(|R||T|)$: Shown in section 4.2
(2) # of mean-field inference in one Q-function computation = 2 (constant): Two embedding steps (Distance embedding, Value embedding) each needs one mean-field inference procedure
(3) # of structure2vec propagation operation in one mean-field inference= $O(|T|^2)$: There is one structure2vec operation from a task to another task and therefore the total number of operations is $|T| \times (|T| - 1)$.
(4) # of neural net computation for each structure2vec propagation operation=C (constant): This is only dependent on the hyperparameter size of neural network and does not increase as number of robots or tasks.
(5) # of neural net computation for inference of random PGM=$O(|T|^2)$ As an offline stage, we infer the semi-clique presence probability for every possible directed edge, i.e. from a task to another task using algorithm introduced in Appendix 6. This algorithm complexity is $O(|T| \times (|T| - 1)) = O(|T|^2)$.

**Training data efficiency.** Training efficiency also is required to obtain scalability. To quantify this we measured the training time required to achieve 93% optimality. As before, we consider a deterministic environment with linear rewards and compare with the exact optimum. Table 4 demonstrates that training time many not necessarily increase with problem size.

Table 4: Training complexity (mean of 20 trials of training, linear & deterministic env.)

| **Linear & Deterministic** | **Testing size : Robot (R) / Task (T)** | | | | | | |
|---|---|---|---|---|---|---|---|
| | 2R/20T | 3R/20T | 3R/30T | 5R/30T | 5R/40T | 8R/40T | 8R/50T |
| Performance with full training | 98.31 | 97.50 | 97.80 | 95.35 | 96.99 | 96.11 | 96.85 |
| # Training for 93 optimality | 19261.2 | 61034.0 | 99032.7 | 48675.3 | 48217.5 | 45360.0 | 47244.2 |

### A.9 CODE FOR THE EXPERIMENT

For the entire codes used for experiments, please go to the following Google drive link for the codes.

