# OpenReview forum: "Embedding a random graph via GNN: mean-field inference theory and RL applications to NP-Hard multi-robot/machine scheduling"
_ICLR.cc/2021/Conference — Reject_

### Official Review · AnonReviewer2 · 2020-10-26
**This paper is well written and easy to follow. The idea is novel and interesting.  Enough theoretical proofs and experimental results are shown to support their arguments.**

**Rating:** 7
**Confidence:** 3

**Review:**

This paper considers multi-robot, multi-task scheduling problems. By introducing the notion of random PGM, authors develop a mean-field inference method and further prove that a modification of a GNN embedding is sufficient to embed a random graph. Experimental results show the near-optimality of the proposed method.

This paper extends the structure2vec algorithm to multi-robot multi-task scheduling problems. I like the idea of formulating the MRRC problem as a sequential decision making problem with graph-structured states. By doing so, the MRRC problem can be solved using random structure2vec with a nearly optimal solution. The proofs seem to be correct and theoretically support the optimality of the proposed method. A reasonable experiment is designed to evaluate the performance of the proposed method.

Overall, this paper is well written. The idea of treating the multi-robot multi-task problem as a sequential decision making problem with directed bipartite graphs is novel.  This new idea naturally transforms the problem into a Bayesian network that can be solved efficiently.
I only have a minor concern regarding the stochastic task completion time. In real cases, it is hard to calculate the exact task completion distribution when tasks are more complex. There is no discussion about this in the paper. And a more complex example should be included in the experiment section.

---

> ### Author Response · Authors · 2020-11-25
> **Some thoughts on stochastic completion times & complex benchmark datasets to be included!**
>
> REVIEWER COMMENT: I only have a minor concern regarding the stochastic task completion time. In real cases, it is hard to calculate the exact task completion distribution when tasks are more complex. There is no discussion about this in the paper.
>
> AUTHOR RESPONSE: Thank you for the comment! It certainly can be a challenge to acquire real distribution in some cases.
> For complex tasks, it can be difficult to know exactly the task time distribution. For such a highly stochastic case, our approach considering assumed probability distribution on task completion time may provide more robust and efficient scheduling results than just using a deterministic estimate of the task completion time.
> In the context of routing and manufacturing scheduling problems, the assumption that we know the distribution of task completion times may make more sense. For example, task completion times in the routing problems can be roughly proportional to the distance between two nodes. In addition, the distribution of task completion time for various manufacturing jobs can be computed from real data.
> To address this concern, we will include the following discussion in the revised paper.
> In some cases, it is reasonable to assume that the task completion time distributions are available. In the context of scheduling for a modern manufacturing system, historical data archives can be mined for the required distributions. In vehicle routing problems, data steams or assumptions such as transportation times being roughly proportional to distance can be employed. However, there are certainly complex systems and tasks where obtaining the distributions is a challenge. In these cases, more robust and efficient scheduling may be obtained with approximate distributions rather than resorting to a simpler deterministic assumption or forgoing the effort entirely.
>
> REVIEWER COMMENT: And a more complex example should be included in the experiment section.
>
> AUTHOR RESPONSE: This is a great suggestion! So we quickly solved a continuous-time, continuous state, time-dependent multi-robot scheduling problem called Minmax (single-depot) Multiple Traveling Salesman Problem (mTSP). While we only have initial results yet, the complete table is to be updated in the anonymized google sheets link: https://docs.google.com/spreadsheets/d/1hkRgloOO1SYM2E2tfKd0cJlUkQ4sFhc42Jq_BhxQtd4/edit?usp=sharing . For minmax mTSP problem, we used the standard dataset provided in (https://profs.info.uaic.ro/~mtsplib/MinMaxMTSP/index.html). As the standard dataset provides the best-up-to-now solution reported for each mTSP problem in the dataset, we compare this result with our method’s result.
>
> In the mTSP problem, a set of homogeneous robots, initially starting from one location called a depot, travels to service a set of spatially distributed tasks. Traveling speed is deterministic. The goal of Minmax mTSP is to minimize the time when all tasks are served, and all robots return to the depot where they were initially located. That is, we want to minimize the maximum length tour. This problem is exactly the same problem as the identical parallel machine scheduling problem (IPMS) except that the initial starting location and final ending location is the same for all robots. As it is a deterministic problem as IPMS is, we don’t have to reassign a robot until it finishes its assigned task. Therefore, new assignment decisions are made only when there is a free robot.

---

### Official Review · AnonReviewer1 · 2020-10-28
**Blind Review**

**Rating:** 6
**Confidence:** 4

**Review:**

This paper studies a class of problems called the Multi-Robot Reward Collection problems. These are scheduling problems that are combinatorial in nature and hard to solve efficiently (in polynomial time). Traditionally, such hard problems have been solved with approximation algorithms or heuristics. Approximation algorithms are useful since they provide theoretical guarantees on the output solution, however their design requires significant problem-specific expertise. On the other hand, heuristics are easy to develop however the solution has no performance guarantees. In contrast to these two approaches, this paper proposes to use graph neural networks (GNN) to solve scheduling problems in graphical settings.

The presented method takes inspiration from structure2vec by Dai et al. (2017), which uses GNNs to compute solutions to several well-known NP-hard problems including the vertex cover and traveling salesperson problems (TSP). Although NP-hard problems are known to be reducible from another problem in the same hardness class, it is not straightforward to extend the solutions to problems involving multiple agents and time scheduling. This paper presents a method to embed graph environments to select actions.

The paper’s main contribution is the probabilistic graphical model for multi-robot multi-task scheduling problems. There are also theoretical and empirical justifications for the method’s performance.

Overall, I think the method will be interesting to the community and the paper is written well except for the points listed below:

- The $(1 - 1/e)$ optimality bound: The approximation factor presented in Theorem 2 is obtained by the greedy maximization over the Q-values. This is computed using the bound on diminishing returns with greedy selection, as presented previously by Nemhauser et al. (1978). Although this result can be used to claim approximation of the optimal Q-values, it is not clear to me how it would translate to performance comparison against the optimal algorithm’s solution. For instance, if the objective function is a minimization problem (not expected return maximization), then we cannot claim that $OPT = Q^N(s, M^*)$ and the approximation to the optimal solution would not hold anymore -- a greedy minimization does not yield $(1 - 1/e)^{-1} = e / (e - 1)$-approximation. I encourage authors to clarify this point.
- The improvement in the computational complexity from $O((R!=T !(R − T )!)^H)$ to $O(|R||T|^3)$ looks impressive. I believe it would be useful to address scalability with the polynomial time runtime of the presented method. For example, can the method work with 100s of robots and tasks -- similar scales were presented before in Dai et al. (2017)? If not, are there any limitations other than the computational resources? I also think that presenting the computation times for each setting will be useful to the reader.
- The experimental evaluations are performed in simulation examples. It would be helpful to show results on real-world data and present generalization performance across different datasets.
- It would be also good to see an explanation of the assumption made in Appendix A.8 with the nonnegative marginal improvement with adding one robot. Can multiple robots occupy the same node of a graph? Is there a notion of traffic in the flow of the graph?
- The method performs auctions in a centralized manner. Any comments on how to make the action selection decentralized would be welcome.

Minor comments:
- Appendix 10 -> Appendix 9
- assigning robot i to task j -> assigning robot i to task p

References:
- Khalil, E., Dai, H., Zhang, Y., Dilkina, B., & Song, L. (2017). Learning combinatorial optimization algorithms over graphs. In Advances in Neural Information Processing Systems (pp. 6348-6358).

- Nemhauser, G. L., Wolsey, L. A., & Fisher, M. L. (1978). An analysis of approximations for maximizing submodular set functions—I. Mathematical programming, 14(1), 265-294.

---

> ### Author Response · Authors · 2020-11-25
> **Optimality bound for minimization, computation, real-world data & decentralization – Response and to be included.**
>
> REVIEWER COMMENT: (The comments on applying Theorem 2 on minimizatiorevin problem)
>
> AUTHOR RESPONSE: Thank you for highlighting this! It is great that you point out this because we did not explicitly mention in our theorem that it is about the maximization problem – we will clarify this in the revision. We will additionally mention how this extends for the minimization case. Since if a given problem is a cost minimization problem, and the marginal value (meaning marginal cost reduction in that context) of adding one robot (or vehicle or machine) diminishes as the number of robots increase, then we can transform the problem into a reward maximization problem by setting a reward = Constant – cost and solve a reward maximization problem. (See the following image for a simplified version of this procedure - https://imgur.com/a/TQFheKR).
>
> REVIEWER COMMENT: (On whether the proposed method can work with 100s of robots and tasks)
>
> AUTHOR RESPONSE: This is an important question. Thank you!
> For the MRRC, the largest problems we solved consisted of 50 tasks and 8 robots. The reason we did not conduct larger experiment on an MRRC problem was not related to the proposed method but rather the data generation complexity (dynamic programming computations are used to determine the distribution of task completion times in the underlying model). In practice, there is no need to generate this dataset to apply the proposed method – one would simply use known distributions for task completion times.
> For the deterministic IPMS problems studied in the appendix we consider larger cases. Table 3 of Appendix A.1.1 demonstrates cases with 100 tasks and 10 machines.  We can and will solve larger problems in the revision and will consider the same scale as Dai (say, 1000 tasks and 50 machines). In addition, we are happy to include computation time data as you advised. Such will be included in the revision.
>
> REVIEWER COMMENT: The experimental evaluations are performed in simulation examples. It would be helpful to show results on real-world data and present generalization performance across different datasets.
>
> AUTHOR RESPONSE: This is a great suggestion! There is one more great question of yours to answer below but the space is limited, so we would be appreciated if you could refer to our answer for the same question made by reviewer3.
>
> REVIEWER COMMENT: The method performs auctions in a centralized manner. Any comments on how to make the action selection decentralized would be welcome.
>
> AUTHOR RESPONSE: We thank this comment for significantly increasing the quality of this paper. Let’s assume that robots can communicate with each other, and all robots are connected in a multi-hop network. Thanks to your comment, the following paragraph and algorithm will be added in the camera-ready version:
>
> (Main text)
> \subsubsection{Decentralized algorithm}. Suppose that robots can communicate with each other and all robots are connected in a multi-hop relaying network.  There is a probability that communication packets can fail, so the impossibility results in asynchronous communication exist. For the classical multi-robot task allocation (MRTA) problem, [Choi et al. 2009] suggests a decentralized solution by showing that some centralized sequentially greedy algorithms can be converted as a decentralized algorithm that in probability achieves a finite time consensus guarantee under this environment. While MRTA is a completely different problem than our assignment decision problem at each timestep, we can use a similar multi-hop relaying method to decentralize our centralized auction-based assignment while maintaining the same performance bound. For details, see Appendix XX.
>
> (In appendix XX)
> \subsection{Decentralization and convergence guarantee} In this section, we show that we can modify the auction procedure in OTAP at each timestep as a special case of \cite{Choi et al. 2009}’s sequential greedy algorithm for solving MRTA problem. This enables us to conclude, without further discussion, that 1) assignment consensus is guaranteed among robots even under frequent communication packet loss and 2) centralized algorithm’s performance bound is inherited.
>
> [Algorithm image] (As this editor does not allow us to use bolded texts to highlight what is different from the centralized algorithm, please see the algorithm at https://imgur.com/a/8dDEbK3.)
>
> Note that, after $n^{th}$ consensus phase, it is guaranteed that all robots will keep sending their message until everyone gets to share common $\mathcal{M}^{(n)}$ which is the same as $\mathcal{M}^{(n)}$ we achieved using the centralized algorithm. Therefore, we get the same assignment result after $N=\max \left(|\mathcal{R}|,\left|T_{t}\right|\right)$ iterations of two phases above.
>
> With this protocol, as time goes to infinity, the probability of achieving consensus converges to one. However, one cannot guarantee that in probability one (=almost surely) it reaches consensus within a finite time.

---

### Official Review · AnonReviewer4 · 2020-10-29
**Tackling a difficult problem; insufficient clarity of the presentation and the significance of the theoretical results**

**Rating:** 5
**Confidence:** 3

**Review:**

#### Summary
The paper considers the problem of multi-robot reward collection (MRRC) in which a number of robots are supposed to perform a number of tasks, in a centralized setting. Finding the optimal scheduling (assigning robots to tasks) under reasonable assignment constraints poses an NP-hard problem. The paper casts this problem as a mean-field inference over random probabilistic graphical models (PGMs) and proposes a two-step hierarchical inference method that employs Q-function estimation for a graph neural network (GNN) representation of MRRC.

#### Strength
1. The paper tackles a very difficult combinatorial problem that is of significant importance in a variety of applications, e.g., in manufacturing and ride-sharing.
2. The proposed method is capable of being trained with a certain number of robots and tasks and then transferred to an instance with a different number of robots and tasks. Therefore, it is, to some extent, generalizable in terms of problem size.
3. The figures in the paper are very illustrative.
4. The authors have made the code available.

#### Weakness
1. The main drawback of the paper is that it lacks clarity. It seems that understanding the paper heavily relies on another paper (Dai et al. 2016).
2. It is hard to understand the theoretical contribution of the paper, e.g., how the local optimality condition in Theorem 1 translates into the overall performance. Also, it is unclear to me whether the order-transferability is guaranteed or not. Since this property is an assumption to the theoretical results regarding the auction-based policy (Lemma 2 and Theorem 2), it is important that it holds.

#### Recommended Decision
Due to insufficient clarity of the presentation and the significance of the theoretical results, I think the paper is borderline in its current form.

#### Supporting Arguments
1. The transition from the first paragraph to the second paragraph in the introduction is abrupt.
2. Please clarify explicitly how the reward function is related to the assignments, states, and actions.
3. It is unclear why certain steps of the algorithm are required. For instance, in the paragraph about structure2vec, some explanation is needed regarding the definition of $\tilde{\mu}_j$ and the particular structure for factorizing the joint distribution (similar to the one in the beginning of Section 4).
4. The proof of Theorem 2 resembles that of the near-optimality result for a greedy algorithm that aims to maximize a submodular function under a cardinality constraint.  Therefore, it may make sense to cite the following paper:
Nemhauser, George L., Laurence A. Wolsey, and Marshall L. Fisher. "An analysis of approximations for maximizing submodular set functions—I." Mathematical programming 14.1 (1978): 265-294.
5. The video in the supplementary material is not mentioned in the paper (nor the appendix).

#### Questions
1. Does the proposed algorithm ensure order-transferability, or does it promote that? Are there any guarantees for being order-transferable?
2. The order-transferability is only tested for the deterministic environment with a linear reward. Does it also hold in the stochastic environment or with a nonlinear reward?
3. What does “minimizing the selection bias” mean in Section 5?

#### Additional Feedback for Improving the Paper
1. It is better to keep the abstract as a single paragraph.
2. The writing is satisfactory. But there are several typos in the paper.
3. Please make the bibliography consistent in terms of details provided for each reference (some references have missing information) and the formatting.
4. To be consistent in the appendix, provide the statements of Lemma 1 and Theorem 1 as well (similar to Lemma 2 and Theorem 2).
5. There are redundant vertical spaces in the appendix.
6. For evaluating the training data efficiency, please clearly denote the unit for computing the training time (e.g., number of steps).

---
#### UPDATE
I thank the authors for their response. My major concerns regarding the paper's clarity and the implications of its theoretical results still hold. Therefore, I would keep my initial score and recommendation.

---

> ### Author Response · Authors · 2020-11-25
> **Order transferability for stochastic/nonlinear cases to be included! Improved clarity on the role of order transferability in the revision.**
>
> REVIEWER COMMENT:  The proof of Theorem 2 resembles that of the near-optimality result for a greedy algorithm that aims to maximize a submodular function under a cardinality constraint. Therefore, it may make sense to cite the following paper: Nemhauser, George L., Laurence A. Wolsey, and Marshall L. Fisher. "An analysis of approximations for maximizing submodular set functions—I." Mathematical programming 14.1 (1978): 265-294.
>
> AUTHOR RESPONSE: Thank you for pointing this out! We did not know where this technique comes from, but thanks to this comment I can cite the right paper. Thank you!
>
> REVIEWER COMMENT: Does the proposed algorithm ensure order-transferability, or does it promote that? Are there any guarantees for being order-transferable?
>
> AUTHOR RESPONSE: The contribution of this paper is limited to suggesting the concept of order-transferability and intuitively justifying the design of a neural network structure to achieve it.  We would explicitly discuss the challenge of theoretical guarantees on order-transferability in a camera-ready version.
>
> In the main text, the following sentence will be added:
> "While the effectiveness of this two-layer graph neural network design was empirically demonstrated, provable theoretical analysis of such composite graph neural networks design is an important topic for subsequent future research."
>
> REVIEWER COMMENT: The order-transferability is only tested for the deterministic environment with a linear reward. Does it also hold in the stochastic environment or with a nonlinear reward?
>
> AUTHOR RESPONSE: Great question! Yes,  it does. We included the results on stochastic environment and nonlinear rewards in our performance results (Table 1). However, we did not include them in our transferability results (Table 2) according to the purpose of each experiment. Based on your advice, we will therefore include those results and provide three more tables on order transferability (Deterministic-Nonlinear setting, Stochastic-Linear setting, Stochastic-Nonlinear setting).
> In the revision we will include such data. (We are creating the graphs.)
>
> REVIEWER COMMENT: What does “minimizing the selection bias” mean in Section 5?
>
> AUTHOR RESPONSE: We meant `When designing our experiments, we focused mainly on preventing the possibility of experimental environment cherry-picking.’ We want our experiments to be representative of a broad class of cases and not simply hand-picked “best case” examples.

---

### Official Review · AnonReviewer3 · 2020-11-01
**The paper presents how to embed a random graph using GNN and shows how  to use this to solve NP-hard scheduling problems with time-varying rewards.**

**Rating:** 7
**Confidence:** 2

**Review:**

The paper presents how to embed a random graph using GNN and shows how  to use this to solve NP-hard scheduling problems for multiple robots. Each state of a Multi-Robot Reward Collection problem is represented as a random probabilistic graphical model.  Then random structure2vec is  used to design a reinforcement learning method that learns near-optimal NP-hard multi-robot scheduling problems with time-dependent rewards.

The core contribution is to show how to solve the scheduling problem using random structure2vec and then design a polynomial time auction algorithm the replace the argmax operation in the order transferability of the Q function.

Experimental results obtained using syntetic data are presented for combinations of tasks and robots in different environments and with different reward functions.  Transferability of learning on different problem sizes and scalability are also presented.

The paper is dense, but it is clearly written and provides background information on the methods proposed.

The work makes a significant contribution to solve a hard problem that has many real applications in manufacturing, ride-sharing, pickup and delivery, etc.

Pros:
1. The problem addressed is important for many real applications.
2. The formulation of the problem and of the proposed solution is presented clearly.
3. The results presented are built on theoretical results.  Some parts require more theoretical developments, but the empirical results support the validity of the approach.

Cons:
1. The figures are too small, very hard to see.
2. The  use of synthetic data for the experimental results is reasonable, but it would have been interesting to use existing data sets (for instance, data sets for vehicle routing problems with temporal constraints or data sets for job shop scheduling) even if not exactly for the same problem to enable comparison and duplication of results.

---

> ### Author Response · Authors · 2020-11-25
> **Figures to be enlarged, minmax mTSP results forthcoming**
>
> REVIEWER COMMENT:
> The figures are too small, very hard to see.
>
> AUTHOR RESPONSE:
> Thank you for pointing this out! We had difficulty to fit our tables into the 8-page limit. As the camera-ready manuscript allows more pages, we will enlarge the figures in the main text.
>
> REVIEWER COMMENT:
> The use of synthetic data for the experimental results is reasonable, but it would have been interesting to use existing data sets (for instance, data sets for vehicle routing problems with temporal constraints or data sets for job shop scheduling) even if not exactly for the same problem to enable comparison and duplication of the result.
>
> AUTHOR RESPONSE: This is a great suggestion! So we quickly solved a continuous-time, continuous state, time-dependent multi-robot scheduling problem called  Minmax (single-depot) Multiple Traveling Salesman Problem  (mTSP). While we only have initial results yet, the complete table is to be updated in the anonymized google sheets link: https://docs.google.com/spreadsheets/d/1hkRgloOO1SYM2E2tfKd0cJlUkQ4sFhc42Jq_BhxQtd4/edit?usp=sharing . For minmax mTSP problem, we used the standard dataset provided in (https://profs.info.uaic.ro/~mtsplib/MinMaxMTSP/index.html). As the standard dataset provides the best-up-to-now solution reported for each mTSP problem in the dataset, we compare this result with our method’s result.
>
> In the mTSP problem, a set of homogeneous robots, initially starting from one location called a depot, travels to service a set of spatially distributed tasks. Traveling speed is deterministic. The goal of Minmax mTSP is to minimize the time when all tasks are served, and all robots return to the depot where they were initially located. That is, we want to minimize the maximum length tour. This problem is exactly the same problem as the identical parallel machine scheduling problem (IPMS) except that the initial starting location and final ending location is the same for all robots. As it is a deterministic problem as IPMS is, we don’t have to reassign a robot until it finishes its assigned task. Therefore, new assignment decisions are made only when there is a free robot.

---

### Decision · Program_Chairs · 2021-01-07
**Final Decision**

**Decision:**

Reject

**Comment:**

This work develops an approach to embed random graphs (some even with dependent edges, hence going beyond classical models such as Erdos-Renyi G(n,p)) using GNNs, and uses these to develop approximation algorithms for solving NP-hard scheduling problems, which typically involve some notion of minimizing weighted completion time (or equivalently, the reward incentivizes early completion, where the age of a job is a linear function of time). This is then used to schedule multiple identical robots  to solve a given set of spatially-distributed tasks. The problems considered---Multi-Robot Reward Collection (MRRC) model vehicle-routing, rideshare etc., and are well-motivated.

This paper takes as motivation earlier work on “structure2vec” by Dai et al. (2016) that uses GNNs to (approximately) solve other NP-hard graph problems: specifically, the random structure2vec developed here is used for an RL approach that learns near-optimal solutions for the MRRC problems considered.

While the paper’s contributions were appreciated in general, its clarity, the fact that the (1 – 1/e) bound of Theorem 2 follows from classical work of Nemhauser et al. (1978), and the fact that real-life examples were not considered, were considered weaknesses. The authors are encouraged to work on these aspects of the paper.